# Exploring regional air pollution transition dynamics: A multi-state markov model approach

Md. Ismail Hossain[1,2], Shuvongkar Sarkar[1*], Md. Injamul Haq Methun[3], Azizur Rahman[4,5,6]

1 Department of Mathematics and Natural Sciences, BRAC University, Dhaka, Bangladesh, 2 Department of Mathematics and Statistics, Mississippi State University, Starkville, Mississippi, United States of America, 3 Department of Statistics, Bangladesh Institute of Governance and Management, Dhaka, Bangladesh, 4 Department of Statistics and Data Science, Jahangirnagar University, Dhaka, Bangladesh, 5 Social Innovation Office, Department of Families, Government of Manitoba, Manitoba, Canada, 6 Department of Community Health Sciences, University of Manitoba, Canada

* sarkarshuvongkar12@gmail.com

## Abstract

### Background

Air pollution, commonly measured by the Air Quality Index (AQI), is a significant global health risk, yet its transition dynamics remain poorly understood. This study aims to investigate the regional air pollution transition dynamics across different air quality states.

### Materials and methods

We analyzed weekly average Air Quality Index (AQI) data from January to September 2024 for 19 countries across Asia, Africa, and Europe, collected from an open-access air quality monitoring platform. According to international standards, AQI was categorized into three states (Good, Unhealthy, Very Unhealthy). We applied a multi-state Markov model to assess weekly transitions between these states and estimate the average time spent in one state before transition.

### Results

Findings indicate that in Asia and Africa, air quality tends to deteriorate more frequently than it improves, with low transition rates from "Very Unhealthy" to better states. Transitions from Unhealthy to Good were less frequent in Asia (HR: 0.09, 95% CI: 0.04, 0.19) and Africa (HR:0.25, 95% CI: 0.11, 0.55) compared to Europe, where air quality showed more stability and improvement. The Good and Unhealthy states in Asia had similar sojourn times of 6.80 (±1.77) and 6.64 (±1.38) weeks, while the Very Unhealthy state lasted 3.36 (±0.98) weeks. The Very Unhealthy state persisted for 0.95 (±0.48) weeks in Africa. Europe maintained the "Good" state longest at 7.68 (±1.98) weeks, with shorter durations for Unhealthy and Very Unhealthy states.

**Data availability statement:** 1. Bangladesh:https://aqicn.org/station/bangladesh-dhaka-department-of-environment/ 2.Indonesia:https://aqicn.org/city/indonesia/kemayoran/ 3.South Korea:https://aqicn.org/city/korea/incheon/ganghwa-gun/ 4.China:https://aqicn.org/city/beijing/ 5.Japan:https://aqicn.org/city/japan/machidashi/machidashinogayamachi/ 6.Thailand:https://aqicn.org/city/bangkok 7.India:https://aqicn.org/city/delhi/mandir-marg/ 8.Kazakhstan:https://aqicn.org/city/kazakhstan/astana/us-embassy/ 9.UAE:https://aqicn.org/city/uae/al-mafraq/ 10.Algeria:https://aqicn.org/city/algeria/algiers/us-embassy/ 11.Ghana:https://aqicn.org/station/ghana-kaneshie-agbogbloshie/ 12.Uganda:https://aqicn.org/city/uganda/kampala/us-embassy/ 13.Bosnia & Herzegovina:https://aqicn.org/city/bosnia-herzegovina/sarajevo/us-embassy/ 14.Portugal:https://aqicn.org/city/portugal/santiago-do-cacem/santiago-do-cacem/ 15.France:https://aqicn.org/city/france/paris/tremblay-en-france/ 16.Spain:https://aqicn.org/city/madrid/ 17.Germany:https://aqicn.org/city/germany/hesse/zierenberg/ 18.UK:https://aqicn.org/city/london/ 19.Poland:https://aqicn.org/city/poland/mazowieckie/warszawa/ursynow/.

**Funding:** The author(s) received no specific funding for this work.

**Competing interests:** The authors have declared that no competing interests exist.

## Conclusion

The study highlights lengthy pollution incidents in Asia and Africa, while Europe demonstrates effective pollution control. These insights can guide policymakers in formulating strategies to mitigate pollution based on regional AQI transition trends.

## Introduction

Air pollution, commonly measured by the air quality index (AQI), is a major global health risk and will remain a pressing public health issue. It causes a range of health problems, such as respiratory and cardiovascular diseases, cancer, and neurological disorders. It is estimated that 9 million premature deaths each year are caused by air pollution [1]. The actual number of deaths related to air pollution is believed to be higher than these estimates since the effects of low-level exposure have yet to be considered in the estimate. There are significant data gaps [2]. To reduce the effects of air pollution, the World Health Organization (WHO) has updated its air quality guidelines and urges countries to take extensive steps to protect public health. Governments worldwide have also implemented various policies to reduce air pollution-related diseases [3].

Despite the efforts of the World Health Organization and governments, of air pollution, about 99% of the world's population still breathes toxic air [4]. Consequently, Exposure to this polluted air can cause various health problems like cardiovascular disease and respiratory illness. The impact on children is severe, as 93% of children breathe toxic air that can cause serious health problems [5]. Comparatively, Asian populations suffer more from air pollution than European populations regarding pollution levels and health impacts. For instance, air pollution significantly contributes to ischemic heart disease mortality in Asia, especially in countries like South Korea and India, where exposure to PM2.5 and household pollutants is high [6]. Furthermore 35% of deaths in Asia were due to air pollution in 2015.The pollution level was recorded as the worst that year. Following this trend, Asia experienced the highest 3.2 million air pollution-related deaths in 2021. Air pollution caused almost 1 million deaths in Africa and 0.57 million in Europe in 2021 [7]. In terms of the economy, the problem of air pollution caused approximately $2.9 trillion in annual costs to the global economy [8].

Numerous studies found that vehicular emissions, construction activities, industrial processes, outdated coal plants, and the burning of fossil fuel for energy were the principal drivers of poor air quality [9–11]. Meteorological Conditions such as temperature [12], wind speed [13], and rainfall [14] are significantly associated with AQI level. Furthermore, AQI is significantly associated with land-use and land-cover and mass concentrations [15–17]. Governments across Asia, Africa, and Europe are implementing a range of strategies to reduce air pollution. Most of the strategies are often tailored to local challenges and resources. For example, developed country of Asia are focusing on expanding renewable energy, and introducing environmental taxes. These strategies have significantly lower carbon emissions and particulate

matter like PM2.5 [18,19]. Whereas, developing countries with dominating informal economies increased government spending on environmental [18,20]. In Africa, most air quality management strategies center on household energy such as cleaner cookstoves [21]. Whereas European countries have taken strong actions to control pollution. They use rules like vehicle emission standards, promote clean fuels, and plan cities better [22,23]. Effective air pollution reduction across all regions requires technical measures, strong governance, cross-sector collaboration, and public engagement [21–23].

Since many of the strategies adopted by different countries are usually tailored to local needs and available resources, it is important to determine the air quality transition time. This is important for several regions. First, it helps policymakers to predict how long populations will be exposed to harmful pollution levels which enables timely advisories and healthcare planning [24]. Understanding transition times also informs the effectiveness of interventions [Renxiao et al., 2022]. Additionally, modeling transition times using tools like Markov chains or machine learning enhances the ability to forecast pollution episodes and manage urban air quality more effectively [25,24]. This knowledge supports designing and adopting effective air quality management plans for different countries [Renxiao et al., 2022, 25]. Thus, it also helps in achieving short-term and long-term emission control targets.

Different researchers use several statistical and machine learning techniques to estimate AQI transition probabilities. For instance, the autoregressive integrated moving average (ARIMA) model was used to predict AQI trends, while it is not efficient in complex atmospheric interactions [26–28] employed a support vector machine model to forecast PM 2.5. The deep learning model was also used to predict AQI [29–31], while it required a large data set to train. [32] used a hybrid deep learning model to forecast changing AQI in China. [33] applied a schematic approach for the spatial-temporal analysis of PM2.5 and AQI in Africa and sub-regions, where AQI was categorized into six categories, and trends were compared with the climate factors, socioeconomic indicators, and terrain features. [34] identified the effect intensity and interaction among the driving factors in China using the geographical detector model, where the primary pollutants are divided into five types of regions based on their spatial patterns.

Despite extensive research, gaps remain in understanding the precise mechanisms behind air pollution progression and its regional disparities. Therefore, this study applies a multi-state Markov model to analyze air quality transition dynamics [35]. Multi-state Markov models have been widely used to investigate disease progression, capturing transitions between different health states such as non-disease, mild, severe, and death [36–40]. Beyond health applications, these models have also been utilized in environmental studies, including the sequential dynamics of rainfall [41], climate and seasonal rainfall variations [42], and ecological state transitions [43]. By applying this approach to Air Quality Index (AQI) transitions, this study aims to provide a quantitative explanation for regional differences and predict future air quality scenarios. Additionally, it will explore how variations in pollution control mechanisms influence long-term air quality trends. These insights will contribute to developing adaptive, proactive, and evidence-based environmental policies, ultimately promoting sustainable air quality improvements and enhancing public health and well-being.

## Materials and methods

### Data source, size, and study design

The data for this study were collected from an open-access online air quality monitoring platform, AQICN (https://aqicn.org/). The dataset includes daily PM2.5 concentration values for 19 countries across three major regions: Asia (Bangladesh, Indonesia, South Korea, China, Japan, Thailand, India, Kazakhstan, UAE), Africa (Algeria, Ghana, Uganda), and Europe (Bosnia & Herzegovina, Portugal, France, Spain, Germany, UK, Poland). Factors such as geographic diversity, data availability, and urban focus guided the selection. This study aimed to include countries from different regions to ensure variability in air quality levels and regional environmental policies. Additionally, one of the primary constraints was the availability of consistent and reliable AQI data. Many countries, particularly in Africa, do not have widely accessible

real-time AQI data. As a result, the number of countries from each region is unequal. To maintain consistency and comparability across countries, we collected AQI data from the capital or a major city, for example, Dhaka for Bangladesh, Paris for France and so on, as these locations typically have better monitoring infrastructure and represent the urban air quality exposure experienced by a large proportion of the population.

The study period spans from January 2024 to September 2024, covering 40 weeks. To derive the Air Quality Index (AQI), the collected PM2.5 values were converted using the standard AQI calculation formula [44]. The AQI conversion formula is given below:

$$I = \frac{I_{HIGH} - I_{LOW}}{C_{HIGH} - C_{LOW}} \times (C - C_{LOW}) + I_{LOW}$$

Here,

$I = AQI$

$C = Concentration\ of\ PM2.5$

$C_{HIGH}, C_{LOW} = Concentration\ breakpoints$

$I_{HIGH}, I_{LOW} = AQI\ breakpoints\ correspomding\ to\ C_{HIGH}, C_{LOW}$

Given the nature of air pollution data, the original dataset was structured as a time series. However, for this study, the data were reformatted into a longitudinal framework by aggregating daily AQI values into weekly averages. This transformation allowed for a more structured analysis of air quality trends over time while reducing short-term fluctuations and measurement noise.

### Study units & framework

Since this study is based on secondary data obtained from an online repository, each country included in the analysis serves as a study analysis unit. The study design treats AQI as a repeated measure over time, with each country's air quality being tracked for 40 consecutive weeks. This preprocessing step served two key purposes. Firstly, daily AQI values are often highly variable due to transient weather conditions or short-term pollution events. Weekly averaging smooths these fluctuations, providing a clearer picture of medium-term air quality trends. Secondly, during the weekly aggregation, any missing daily values within a given week were handled by computing the average of the available days. In cases where at least four or more valid daily observations were present in a week, the weekly average was retained to ensure reliability. Weeks with excessive missingness (fewer than four valid daily observations) were imputed using a custom interpolation function to ensure data quality and completeness for the final analysis. The detailed code is available upon request.

By structuring the dataset longitudinally, each country is represented by 40 distinct time points, allowing for a more comprehensive examination of how air quality evolves across different regions. This approach ensures that regional air quality patterns and trends can be meaningfully analyzed within the study period.

### Variables

The primary variable of interest in this study is the Air Quality Index (AQI), which has been categorized into three discrete states based on internationally recognized AQI classifications (https://aqicn.org/scale/):

$$AQI = \begin{cases} \text{Good, If AQI} \leq 150 \\ \text{Unhealthy, If } 150 < \text{AQI} \leq 200 \\ \text{Very Unhealthy, If AQI} > 200 \end{cases}$$

Here, "Good" state represents low pollution levels with minimal health risks. "Unhealthy" state represents moderate to high pollution levels that may pose significant health risks, particularly for vulnerable populations. "Very Unhealthy" state represents severe pollution levels that pose serious health threats to the general population.

The key covariate used in this study is the regional classification of the 19 selected countries. Based on geographical location, the countries are categorized into three regions: Asia, Africa, and Europe. This regional classification serves as an essential factor in the analysis, allowing for comparisons between regions and an examination of how regional differences influence AQI transitions.

## Statistical analysis

Graphical methods were employed to describe the dataset effectively, particularly line graphs, to illustrate trends in air quality over time. Weekly average AQI values were visualized at the country level and the region level (Asia, Africa, and Europe). These visual representations provide an overview of variations in air quality across different locations over the study period, facilitating an intuitive understanding of regional disparities and temporal fluctuations.

A Multi-State Markov Model (MSM) was used to model the transitions between air quality states. This model is well-suited for analyzing time-dependent categorical data where subjects transition between multiple discrete states. This study defined three AQI states: Good Air Quality (State 1), Unhealthy Air Quality (State 2), and Very Unhealthy Air Quality (State 3). A transition occurs when the AQI status of a country shifts from one state to another between consecutive time points. The model allows for estimating the likelihood of moving from one state to another while accounting for time-dependent changes in air quality. If a state permits further transitions, it is considered a transient state, whereas an absorbing state would indicate a permanent condition with no further transitions. However, in this study, all states were assumed to be transient, as air quality can deteriorate or improve over time.

Nine possible transitions between the three states were considered, as illustrated in Fig 1. The transition probabilities were estimated using hazard ratios (HRs) with corresponding 95% confidence intervals (CIs), derived from model parameters estimated through maximum likelihood estimation (MLE). These estimates quantify the relative risk of moving from one air quality category to another across different regions. For a detailed explanation of the Multi-State Markov Model (MSMM) methodology and its underlying mechanism, refer to [45].

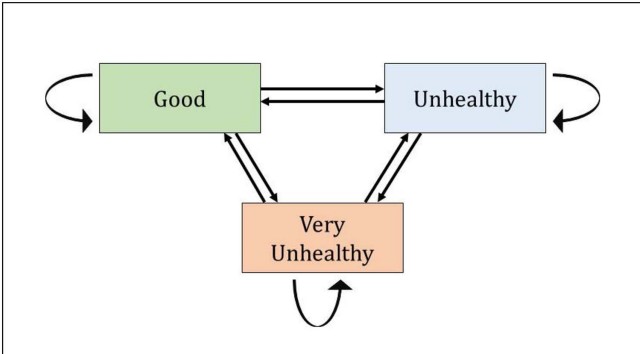

Source: Author's conceptualization

**Fig 1. The three AQI state transitions in the Markov model.**

To address model assumptions, the MSM approach assumes a Markovian process where future states depend only on the current state. Given the use of secondary weekly-aggregated AQI data, some transition timing variability may be missed. Despite these limitations, the model offers meaningful insights into regional air quality dynamics. Additionally, the identical transition intensity matrices observed in both early and late periods support the time-homogeneous assumption, indicating that transition rates between air quality states remain stable over time.

## Software & tools

In this study, data preparation and analysis were conducted using R programming, incorporating several specialized packages to ensure accuracy and efficiency. The "msm" package was utilized for implementing the Multi-State Markov Model, enabling robust estimation of transition probabilities and model parameters [45]. Custom functions were developed to process and structure the dataset appropriately to facilitate data preparation. The complete script for data preprocessing is available upon request.

## Results

Fig 2 presents the weekly average AQI trends across nine Asian countries over a 40-week period, highlighting fluctuations in air quality over time. While most countries generally maintained moderate AQI levels, Bangladesh and India experienced very unhealthy air quality during the first 20 weeks. In contrast, Japan sustained healthy air quality throughout the entire 40-week span. These results underscore that Asia consistently records the highest AQI levels, reflecting ongoing and persistent air pollution challenges in the region.

African Countries shows considerable fluctuations, with a notable decline in AQI after the initial weeks, followed by irregular peaks (Fig 3). Ghana shows occasional spikes into the very unhealthy range, whereas Algeria and Uganda generally fluctuate between good and unhealthy categories, with Uganda displaying more variability.

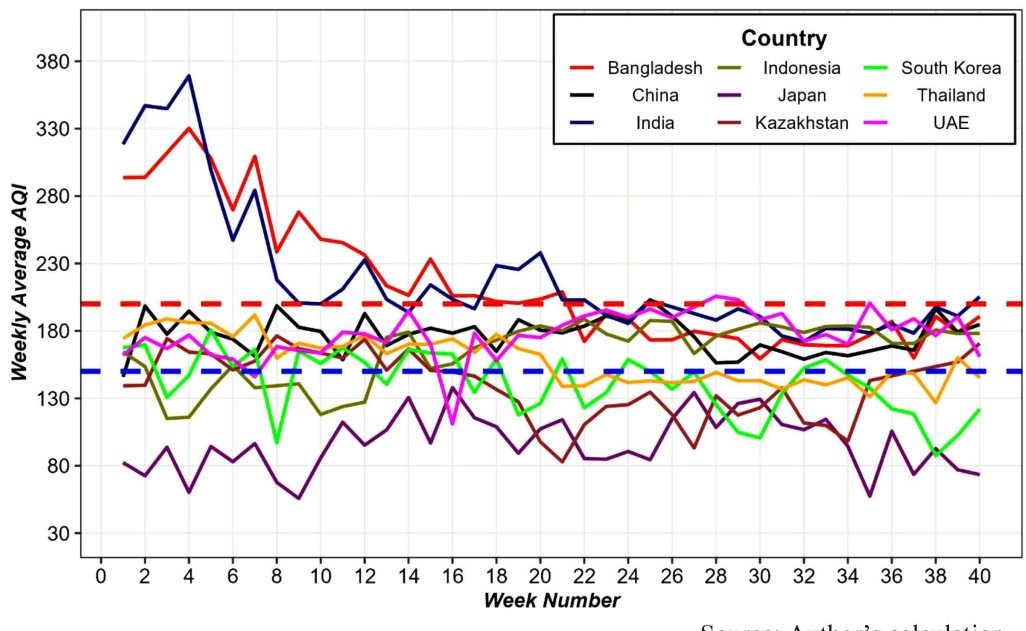

Source: Author's calculation

**Fig 2. Weekly average AQI by Asian Countries.** The region above the red dotted horizontal line indicates very unhealthy air quality (AQI ≥ 200), the region between the blue and red dotted horizontal lines indicates unhealthy air quality, and the region below the blue line indicates good air quality (AQI ≤ 150).

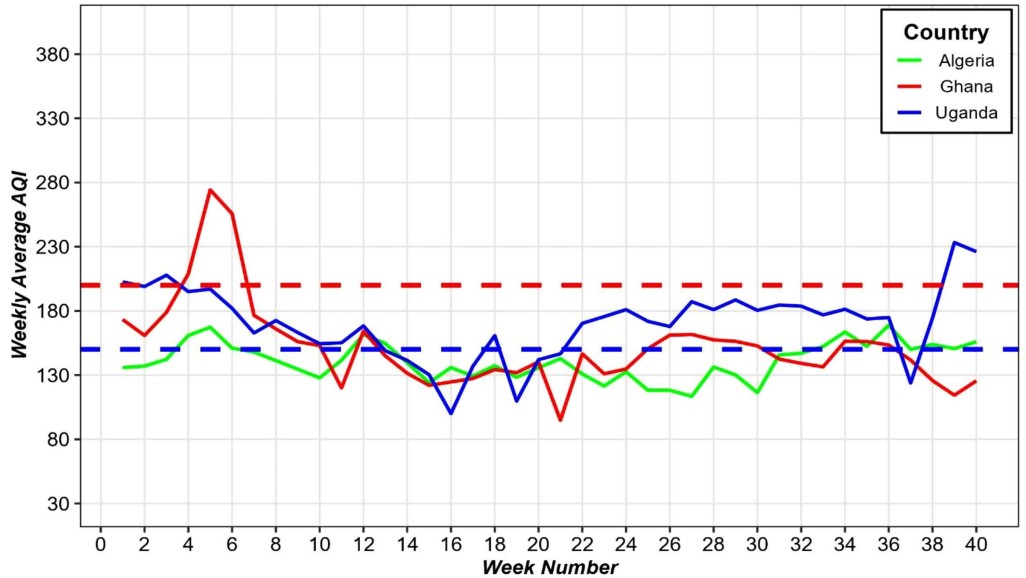

Source: Author's calculation

**Fig 3. Weekly average AQI by African Countries.** The region above the red dotted horizontal line indicates **very unhealthy air quality (AQI ≥ 200)**, the region between the blue and red dotted horizontal lines indicates **unhealthy air quality**, and the region below the blue line indicates **good air quality (AQI ≤ 150)**.

Across Europe, air quality generally remains at comparatively lower AQI levels, indicating a more stable and controlled pattern (Fig 4). In contrast, Bosnia and Herzegovina presents a notable deviation, with air quality persisting at very unhealthy levels at fourth week before showing steady improvement and ultimately reaching a healthy range within ten weeks.

Table 1 outlines the transitions between the three AQI categories: Good, Unhealthy, and Very Unhealthy, from one week to the next. A majority of the measurements remained in the same AQI category week-to-week. When air quality was initially categorized as "Good," it largely (89.1% time) stayed that way in the following week. However, it was observed that some transitions: 10.2% (41 measurements) moved to "Unhealthy," and 0.7% (3 measurements) became "Very Unhealthy". Among those initially recorded as "Unhealthy", 218 (78.4%) remained in the "Unhealthy" category, 44 (15.8%) improved to "Good", and 16 (5.8%) worsened to "Very Unhealthy". For measurements initially categorized as "Very Unhealthy", 40 (66.7%) stayed in the same category, while 18 (30.0%) improved to "Unhealthy", and only 2 (3.3%) improved to "Good".

Fig 5 illustrates AQI state transitions over 1-week, 2-week, 3-week, and 4-week periods for Asia. Figs 6 and 7 depict similar information for Europe and Africa. The transition diagrams show how air quality changes over time using a Multi-State Markov Model. The results reveal a high probability of AQI staying in the same state week-to-week across all regions. However, in Asia and Africa, air pollution tends to worsen more often than it improves, with a low transition rate from "Very Unhealthy" to better states, indicating prolonged pollution episodes. In contrast, Europe displays a different pattern, with more stable or improving air quality trends than Asia and Africa.

In Asia, the expected sojourn time is longest in the "Good" state, averaging 6.80 (±1.77) weeks, followed closely by the "Unhealthy" state at 6.64 (±1.38) weeks. The "Very Unhealthy" state has the shortest duration, with an average of 3.36 (±0.98) weeks. In Africa, the sojourn times are generally shorter. The "Good" state lasts around 4.57 (±1.49) weeks, while the "Unhealthy" state persists for 2.66 (±0.82) weeks. The "Very Unhealthy" state has the briefest stay, averaging only 0.95 (±0.48) weeks. In Europe, the "Good" state has the highest sojourn time at 7.68 (±1.98) weeks, but the "Unhealthy"

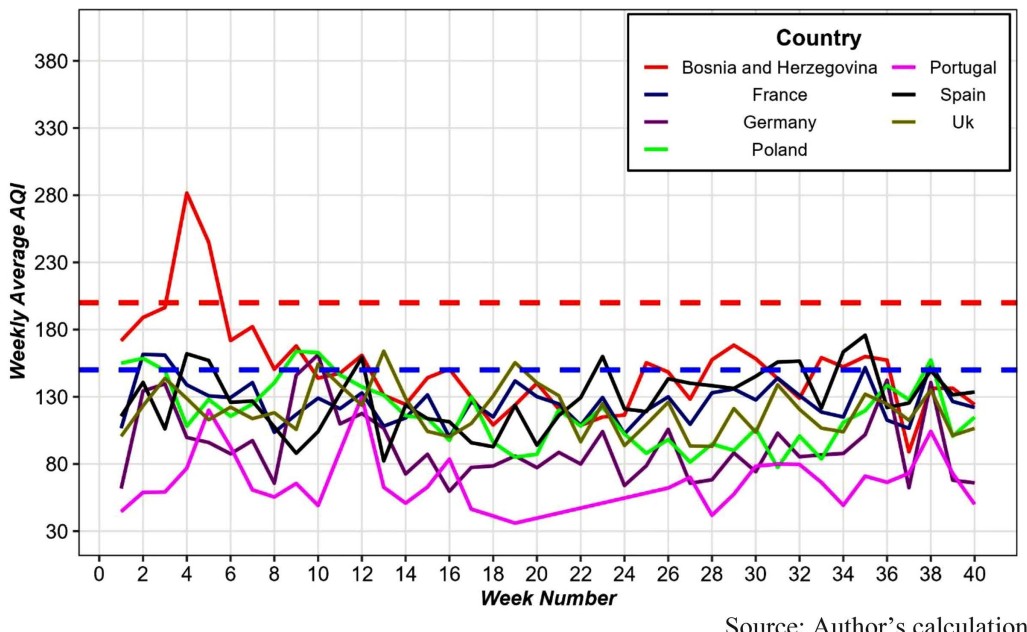

Source: Author's calculation

**Fig 4. Weekly average AQI by European Countries.** The region above the red dotted horizontal line indicates **very unhealthy air quality (AQI ≥ 200)**, the region between the blue and red dotted horizontal lines indicates **unhealthy air quality**, and the region below the blue line indicates **good air quality (AQI ≤ 150)**.

**Table 1. Observed number of transitions from one week to the following week.**

| | | | | |
|---|---|---|---|---|
| *Good* | 359 (89.1%) | 41 (10.2%) | 3 (0.7%) | **403** |
| *Unhealthy* | 44 (15.8%) | 218 (78.4%) | 16 (5.8%) | **278** |
| *Very unhealthy* | 2 (3.3%) | 18 (30.0%) | 40 (66.7%) | **60** |

Source: Author's calculation.

and "Very Unhealthy" states have significantly shorter durations, averaging just 0.90 (±0.22) weeks and 0.68 (±0.46) weeks, respectively.

Table 2 examines the influence of regional covariates on weekly transitions between AQI categories (Good, Unhealthy, and Very Unhealthy) over the study period. Hazard ratios (HRs) with corresponding 95% confidence intervals (CIs) are provided for transitions involving Asia and Africa, with Europe as the reference category. Notably, certain transitions exhibit large CIs, reflecting the rarity of extreme AQI shifts, such as from Good to Very Unhealthy or vice versa. These patterns align with real-world scenarios where sudden drastic changes in air quality are uncommon.

The likelihood of transitioning from good to unhealthy showed slight regional differences. Asia had an HR of 1.10 (95% CI: 0.52, 2.34), while Africa had an HR of 1.59 (95% CI: 0.64, 3.95), both suggesting slightly higher risks than Europe. In contrast, transitions from good to very unhealthy were extremely rare across all regions, resulting in large CIs (e.g., 2.99 [95% CI: 0.00, 1.37e+04] for Asia and 7.69 [95% CI: 0.00, 5.46e+04] for Africa), consistent with the low frequency of such drastic air quality deteriorations. Transitions from unhealthy to good occurred less frequently in Asia (HR: 0.09, 95% CI: 0.04, 0.19) and Africa (HR: 0.25, 95% CI: 0.11, 0.55) compared to Europe, highlighting slower recovery to better air quality in these regions. Conversely, the likelihood of shifting from unhealthy to very unhealthy was relatively low in both

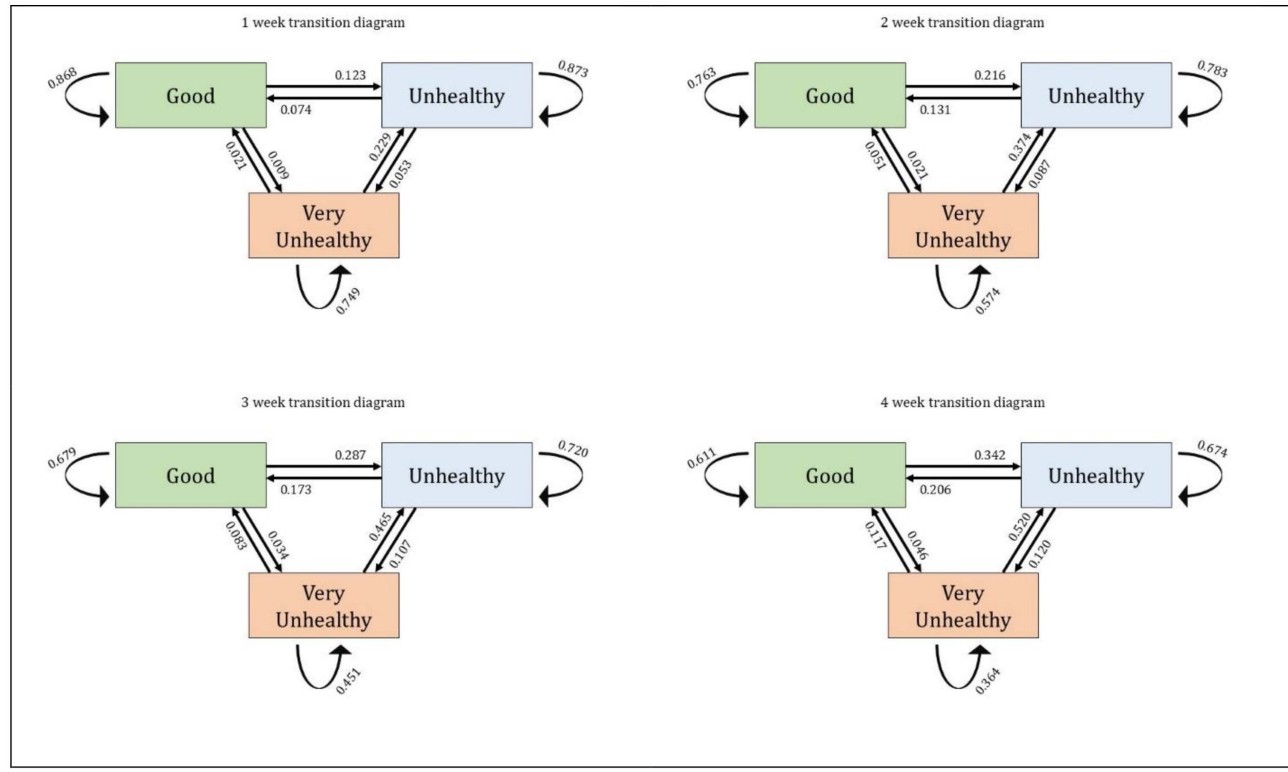

Source: Author's calculation

**Fig 5. State transition diagrams for Asia region countries.**

Asia (HR: 0.40, 95% CI: 0.07, 2.21) and Africa (HR: 0.89, 95% CI: 0.12, 6.38), though the wide intervals reflect uncertainty due to limited observations. Transitions from very unhealthy to good were exceedingly rare. The HRs for Asia (1.62, 95% CI: 0.00, 5.72e + 16) and Africa (0.37, 95% CI: 0.00, 4.39e + 17) underscore the infrequency of such drastic improvements. Transitions from very unhealthy to unhealthy were less likely in Asia (HR: 0.20, 95% CI: 0.04, 0.87), while Africa's HR of 0.73 (95% CI: 0.14, 3.93) was closer to Europe.

Fig 8 illustrates the transition probabilities of three AQI categories, Good, Unhealthy, and Very Unhealthy, over a 10-week period across three regions: Africa, Asia, and Europe. Transition probability refers to the likelihood of remaining in a given AQI category without transitioning to another state during the observed weeks. The curves highlight variations in air quality stability across regions and AQI states.

For the good category, the transition probability starts high in all regions but declines at different rates. Europe exhibits the most stable good air quality, with a gradual decline in transition probability over time. In contrast, Asia and Africa experience more rapid declines, with Africa showing the steepest drop, indicating that good air quality is less persistent in these regions. In the unhealthy category, the transition probability is moderate across regions, but the patterns differ. Europe shows the fastest decline, suggesting quicker transitions out of the unhealthy state, likely improving to good. Asia and Africa exhibit slower declines, indicating that unhealthy air quality persists longer in these regions compared to Europe. The very unhealthy category demonstrates the steepest overall declines in transition probabilities. Africa and Europe show sharp decreases within the first few weeks, reflecting the short-lived nature of very unhealthy air quality in these regions. However, in Asia, the transition probability of the very unhealthy state decreases more gradually, suggesting that this extreme AQI condition tends to persist longer in Asian regions.

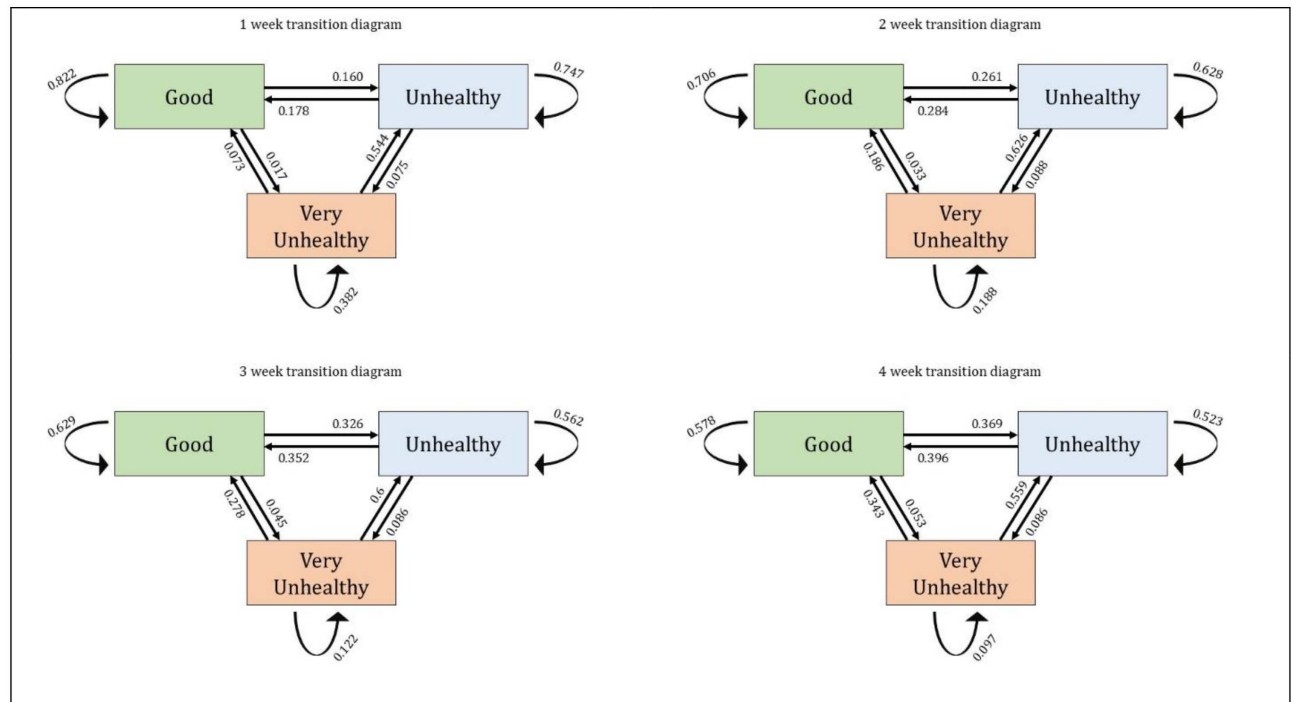

Source: Author's calculation

**Fig 6. State transition diagrams for Africa region countries.**

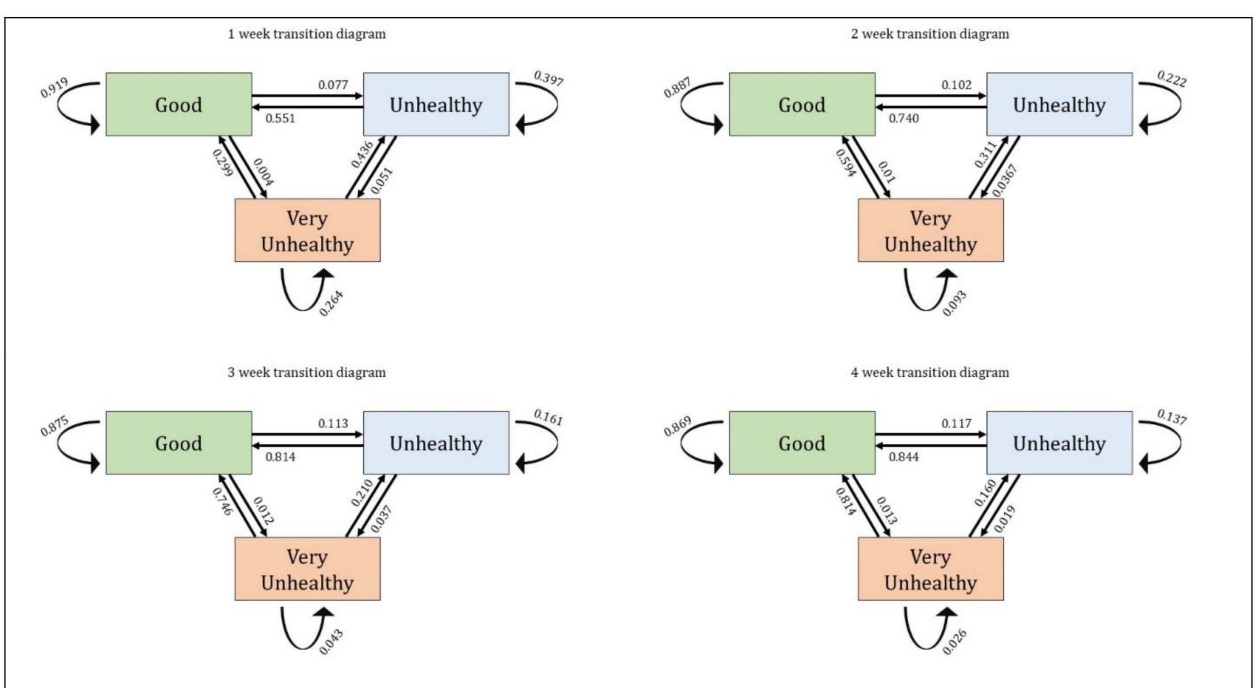

Source: Author's calculation

**Fig 7. State transition diagrams for Europe region countries.**

**Table 2. Effects of regions on transitions between weekly AQI categories.**

| Variable | Good-Unhealthy | Good-Very Unhealthy | Unhealthy-Good | Unhealthy-Very Unhealthy | Very Unhealthy-Good | Very Unhealthy-Unhealthy |
|---|---|---|---|---|---|---|
| | HR (95% CI) | HR (95% CI) | HR (95% CI) | HR (95% CI) | HR (95% CI) | HR (95% CI) |
| **Region** | | | | | | |
| Asia | 1.10 (0.52, 2.34) | 2.99 (0.00, 1.37e+04) | 0.09 (0.04, 0.19) | 0.40 (0.07, 2.21) | 1.62 (0.00, 5.72e+16) | 0.20 (0.04, 0.87) |
| Africa | 1.59 (0.64, 3.95) | 7.69 (0.00, 5.46e+04) | 0.25 (0.11, 0.55) | 0.89 (0.12, 6.38) | 0.37 (0.00, 4.39e+17) | 0.73 (0.14, 3.93) |
| Europe (Ref.) | | | | | | |

Source: Author's calculation.

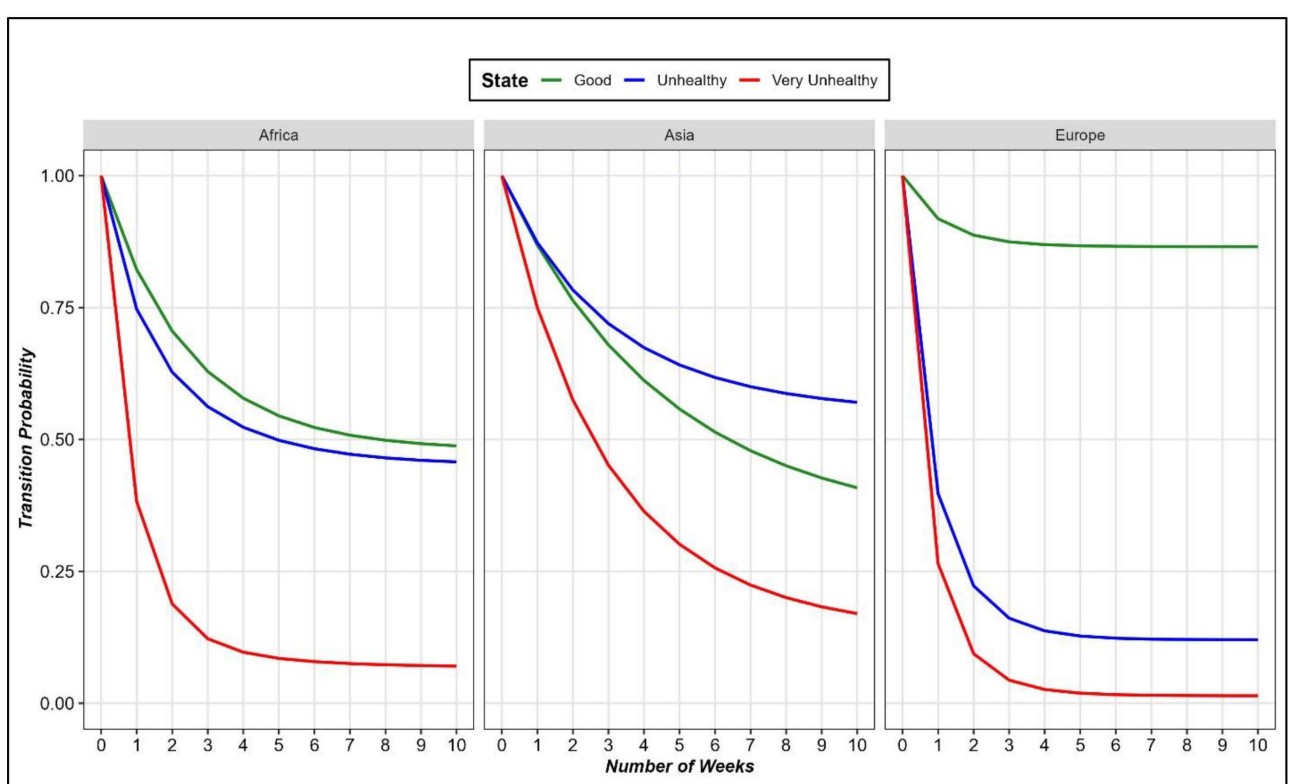

Source: Author's calculation

**Fig 8. Transition probability of weekly AQI categories across regions.**

## Discussion

One of the biggest environmental problems today is outdoor air pollution [46]. It has been proven in various studies that the Earth's atmosphere is continuously being polluted due to multiple factors, such as industrialization, urbanization, and excessive use of nitrogen-based fertilizers [47–50]. As a result, 4.2 million to 7 million deaths occur annually due to outdoor air pollution [51].

When comparing different regions of the world, air pollution is observed globally, but it is significantly higher (approximately 80 percent of total who are directly exposed to unsafe PM2.5 concentration) in low- and middle-income countries

[52]. In developing countries like Bangladesh [53], India [54], Pakistan [55], and even China [56], air pollution has become a serious threat to both climate and public health. A recent study in Bangladesh found that the annual average concentration of PM2.5 in Dhaka was nearly 18 times higher than the World Health Organization's recommended threshold, highlighting the severity of air pollution in the region [57]. In the African region, a recent study estimated that 1.1 million deaths occurred in 2019 alone due to poor air quality [58]. In contrast, European countries maintain better air quality through strict regulations and technological advancements, though many still need stronger policies for further improvement [59].

It is evident that there is a regional disparity in air quality, and one of the key objectives of this study was to provide a mathematical explanation of this difference and predict possible future air quality scenarios. Using the Multi-State Markov Model (MSMM), we analyzed AQI state transition, which helps understand how air quality changes over time in different regions. Another goal of our study was to highlight the variations in regional control mechanisms and propose appropriate policies for different regions. Since Europe has comparatively better controlled air pollution, we considered this region as a reference and explored how MSMM-based predictions can help identify long-term air quality improvement strategies for Asia and Africa.

According to our findings, AQI state transition analysis was conducted for 19 countries from three continental regions (Asia, Africa, and Europe) over 40 weeks from January to September 2024. A total of 760 AQI measurements were analyzed, and results showed that in most cases, air quality remained unchanged from one week to the next. Our study found that in Asia and Africa, air pollution tends to deteriorate more often than it improves. The transition from "Very Unhealthy" to a better category was found to be relatively low, meaning that once pollution worsens in these regions, it remains at high levels for an extended period and does not improve easily through natural processes. Previous studies have also indicated that regions with high levels of particulate matter (PM2.5) and nitrogen dioxide (NO2) emissions experience extended periods of poor air quality [60]. In contrast, Europe presented a completely different pattern. The AQI in European countries remains more stable in the better categories, and the transition from polluted to improved conditions is comparatively higher. This indicates that European air pollution control policies and environmental management systems are significantly more effective than those in Asia and Africa.

Another important aspect observed in this study is the relatively low likelihood of drastic shifts in air quality, particularly sudden transitions from clean to heavily polluted air or vice versa. This aligns with real-world patterns where air pollution levels typically fluctuate gradually rather than experiencing sudden changes [61], except in cases of extreme environmental disturbances such as wildfires, acid rain, global warming etc. [62]. This stability in air quality patterns suggests that while pollution control measures in some regions may be insufficient to achieve rapid improvements, complete environmental collapses due to air quality shifts remain rare.

Additionally, a 10-week transition trend analysis was conducted, which shows the probability of AQI remaining unchanged in a given category, meaning the likelihood of not transitioning to another state. For the "Good" category, European countries showed the highest stability, where air quality gradually declines over time. However, in Asia and Africa, deterioration occurs more rapidly, especially in African countries, where the decline is the sharpest, indicating that good air quality does not persist for long in this region. For the "Unhealthy" category, transition probability decreases the fastest in Europe, suggesting a quicker transition to better air quality. However, in Asia and Africa, the "Unhealthy" state tends to persist for a longer period, indicating weak pollution control measures in these regions. For the "Very Unhealthy" category, Africa and Europe experience a rapid decline, meaning extremely poor air quality does not last for long in these regions. However, in Asia, this decline occurs at a much slower rate, indicating that severely polluted air remains for an extended period in this region.

## Strengths and limitations of the study

This study provides a comprehensive analysis of air quality transitions across three major regions (Asia, Africa, and Europe) using a Multi-State Markov Model (MSMM). One of its key strengths lies in its ability to capture weekly variations

in AQI over a 40-week period, allowing for a dynamic assessment of pollution trends rather than relying on static observations. The inclusion of multiple regions enables a comparative approach, where Europe's more effective pollution control measures serve as a benchmark for understanding the persistent challenges in Asia and Africa. By identifying patterns of air quality deterioration and improvement over time, the study offers valuable insights that can guide future policy interventions.

Despite these strengths, the study has certain limitations. A significant constraint is the limited inclusion of covariates, as only the region was considered in the analysis due to data availability. Various important factors, such as industrial activities, population density, meteorological conditions, and environmental policies, were not incorporated, which could have influenced AQI transitions. Additionally, the study assumes homogeneity within regions, meaning it does not account for country-specific variations in pollution levels, enforcement of environmental laws, or economic conditions. The dataset, covering 19 countries, provides meaningful insights but may not fully capture global pollution dynamics. By focusing on capital or major cities, our analysis may not capture intra-country variations in air quality, particularly between urban and rural areas. Due to data availability constraints, the sample includes more countries from Asia and Europe compared to Africa. This could introduce regional imbalance and limit the generalizability of region-specific comparisons. Regarding seasonality, while we recognize that AQI exhibits seasonal patterns, the structure of our analysis (monthly average transitions across multiple years) helps smooth short-term fluctuations. Moreover, our primary objective was to evaluate long-term transition dynamics across broad regions using a parsimonious Multi-State Markov Model. Future studies with richer datasets could build upon this framework by incorporating meteorological covariates and explicitly modeling seasonal effects (e.g., via time-varying covariates or seasonal transition matrices). In addition, some transitions, particularly from "Good" to "Very Unhealthy" and from "Very Unhealthy" to "Good", were extremely rare in the dataset. As a result, the corresponding hazard ratio estimates in these transition groups have extremely wide confidence intervals, suggesting limited precision and potential numerical instability in those specific model estimates. These rare events should be interpreted with caution, and future studies with larger or longer-term datasets may help improve the reliability of such estimates. Another limitation of this study is the Markov assumption that the next state depends only on the current state. Our tests indicate some influence from earlier states, meaning the process is not fully Markovian. While the MSMM offers useful insights, future work should explore models that capture more complex temporal dependencies.

Despite these limitations, the study provides a strong foundation for understanding regional disparities in air quality and pollution control effectiveness. Future research should aim to include a broader range of covariates, expand the dataset, and explore sector-specific pollution sources to refine the accuracy and applicability of findings.

## Conclusion

This study highlights that air pollution is a significant regional challenge, with Asia and Africa experiencing prolonged periods of poor air quality, while Europe demonstrates more effective pollution control through stronger policies. Using the Multi-State Markov Model (MSMM) to analyze weekly AQI transitions, we found that transitions from good to unhealthy air quality are more frequent in Asia and Africa than in Europe, and recovery to better air quality is slower in these regions. Sudden drastic changes in AQI were rare across all regions. These results suggest that adopting European-style pollution control measures, such as stricter emission standards, increased renewable energy use, improved public transportation, enhanced air quality monitoring, and expanded urban green spaces, could benefit Asia and Africa. Our findings provide valuable insights for policymakers, helping them develop targeted strategies to mitigate air pollution. Future MSMM-based predictions will further enhance understanding of air quality dynamics and support long-term environmental planning.

## Supporting information

**S1 Fig. Transition probability of weekly AQI categories across 19 countries.**
(DOCX)

## Acknowledgments

The authors thank the World Air Quality Index team for allowing us to use air quality data for this study.

## Author contributions

**Conceptualization:** Md. Ismail Hossain, Azizur Rahman.

**Data curation:** Md. Ismail Hossain.

**Formal analysis:** Md. Ismail Hossain.

**Methodology:** Md. Ismail Hossain.

**Resources:** Md. Ismail Hossain.

**Software:** Md. Ismail Hossain, Shuvongkar Sarkar.

**Supervision:** Azizur Rahman.

**Validation:** Md. Ismail Hossain.

**Writing – original draft:** Md. Ismail Hossain, Shuvongkar Sarkar, Md. Injamul Haq Methun.

**Writing – review & editing:** Md. Ismail Hossain, Shuvongkar Sarkar, Md. Injamul Haq Methun, Azizur Rahman.

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
