## [Decision Letter · Decision Letter 0]

18 Jun 2025

PLOS ONE

Dear Dr. Sarkar,

Referring to the PLOS ONE editorial criteria for publication, submitted manuscripts should be either scientifically valid or technically sound, not solely based on the proposed importance or significance of their study. Failure to meet these criteria will result in the rejection of all submitted manuscripts. If requesting a revision, to avoid multiple rounds of revisions, please give clear and constructive responses to reviewers’ advice and prepare the revised manuscript so that it's ready for acceptance. 

After careful consideration, we feel that your manuscript has merit but does not fully meet PLOS ONE's publication criteria as it currently stands. Therefore, we invite you to submit a revised version of the manuscript that addresses the points raised during the review process.

We look forward to receiving your revised manuscript.

Kind regards,

Assoc. Prof. Phakkharawat Sittiprapaporn, Ph.D.

Academic Editor

PLOS ONE

Journal Requirements:

3. Thank you for stating the following in your Competing Interests section: [N/A]

4. Thank you for uploading your study's underlying data set. Unfortunately, the repository you have noted in your Data Availability statement does not qualify as an acceptable data repository according to PLOS's standards.

Additional Editor Comments:

Reviewers' comments:

Reviewer's Responses to Questions

**Comments to the Author**

1. Is the manuscript technically sound, and do the data support the conclusions?

Reviewer #1: Partly

Reviewer #2: Partly

Reviewer #3: Yes

Reviewer #4: Partly

Reviewer #5: Yes

Reviewer #6: Partly

2. Has the statistical analysis been performed appropriately and rigorously?

Reviewer #1: Yes

Reviewer #2: Yes

Reviewer #3: Yes

Reviewer #4: Yes

Reviewer #5: Yes

Reviewer #6: No

3. Have the authors made all data underlying the findings in their manuscript fully available?

Reviewer #1: Yes

Reviewer #2: Yes

Reviewer #3: No

Reviewer #4: Yes

Reviewer #5: Yes

Reviewer #6: Yes

4. Is the manuscript presented in an intelligible fashion and written in standard English?

Reviewer #1: Yes

Reviewer #2: Yes

Reviewer #3: Yes

Reviewer #4: Yes

Reviewer #5: Yes

Reviewer #6: Yes

Reviewer #1: Expand on the methodology, particularly regarding model assumptions, diagnostics, and any limitations of the Markov approach.

Clarify the selection process for countries and discuss any potential biases or limitations in the AQI data.

Provide more detail on data preprocessing and handling of missing values, if applicable.

Improve the English language and presentation by correcting grammatical errors and refining awkward sentences.

Consider including additional visualizations (e.g., transition probability matrices, state occupancy plots) to aid in the interpretation of results.

Discuss the policy implications of findings in greater depth, particularly how transition dynamics can inform targeted interventions.

Reviewer #2: General Assessment. The manuscript is generally well written and presents an application of a multi-state Markov model to analyze air quality transitions across regions. The topic is important, and the modeling approach is appropriate. However, the analysis lacks depth and granularity in several key areas, which limits the interpretability and policy relevance of the findings.

Major Comments

Regional Classification Ambiguity. The manuscript states that countries were grouped into Asia, Africa, and Europe based on both geographical location and economic development levels. However, it is unclear how economic development was actually incorporated into this classification. The grouping appears to be purely geographical. Given that the authors themselves highlight the prevalence of unsafe PM2.5 concentrations in low- and middle-income countries, it would be more appropriate to stratify the analysis by income level (e.g., low-, middle-, and high-income countries) or at least include this as a covariate.

Justification for Time Intervals. The rationale for selecting 1-, 2-, 3-, 4-, and 10-week intervals for tracking transition probabilities is not provided. Why were these specific intervals chosen? Are they based on prior literature, empirical patterns, or policy relevance? Clarifying this would strengthen the methodological transparency.

Seasonality and Covariates. The manuscript does not address clear seasonal patterns in the AQI data. This omission is critical, as air pollution is known to vary with meteorological conditions such as temperature, humidity, and wind. Including covariates such as average outdoor temperature, CO emissions, or precipitation would significantly enhance the explanatory power of the model. The current analysis, which only includes region as a covariate, is too limited and makes the study feel thin.

Health Outcomes Missing. Although the introduction and discussion emphasize the health impacts of air pollution, the study does not attempt to link AQI transitions to health outcomes. If hospitalization or morbidity data are unavailable, even a discussion of potential proxies or future directions would be helpful. Weekly or seasonal hospitalization rates, if accessible, could provide valuable context.

Mobility and Behavioral Data. The authors may consider incorporating or at least discussing the potential use of mobility data (e.g., Google Community Mobility Reports) to contextualize changes in air quality, especially during periods of lockdown or reduced activity. While these reports are no longer updated post-2022, they could still be useful for historical comparison or model validation.

European “Success” Requires Caution. The conclusion that Europe has been more successful in controlling pollution is not sufficiently substantiated. The analysis is sensitive to how countries are grouped. A small change in country composition could significantly alter the results. This limitation should be acknowledged more explicitly.

Minor Suggestions

Improve the clarity of figures, especially transition diagrams, which are currently difficult to interpret without more detailed legends or annotations.

Consider including a table summarizing country-level characteristics (e.g., GDP, population, urbanization rate) to contextualize the findings.

Reviewer #3: as each emprical study , there are two types of reamarks ; about strengthnes of the study , we find :

-Data and statistics used in the study are recen

- It is a very interesting thematic topic related to the SDG, which is usually explored and discussed by authors given the importance of subjects such as pollution and global health.

- A well selected econometric tool.

But there are some Shrtcomings .

Reviewer #4: Thank you for the opportunity to review this manuscript concerning the transition dynamics of air pollution.

Although the quantitative approach of the manuscript is timely, there are points in the methodology that are problematic and that the author(s) should address, revise, and/or consider when taking this work forward. These topics are summarized in list format below.

- Justify the chosen study period, indicating the resulting weaknesses. For example, in addition to seasonality, it is not possible to assess environmental effects.

- The chosen observation unit (country) is a very broad aggregate that can substantially distort the results. It is appropriate for the study to be carried out at the city or metropolitan region level, for example.

- The methodology should specify how possible interactions between countries are treated by the model. Or indicate this topic as a weakness.

- [149 – 150] “Based on geographical location and economic development levels, the countries are categorized into three regions: Asia, Africa, and Europe.” – insert reference for this categorization. It is not clear how “economic development levels” defines region.

- Still regarding the region variable, in the text this variable is treated as a covariate, however it is not an input of the adopted model and I suggest the exclusion of the term covariate.

Reviewer #5: This manuscript analyzed the weekly average Air Quality Index (AQI) in 19 countries across Asia, Africa, and Europe, using data collected from an open-access air quality monitoring platform. The authors did a commendable job of demonstrating how Markov models can be applied in the monitoring or surveillance of AQI, and potentially even in predicting air quality trends. The manuscript is generally well organized, written in clear English, and engaging. The topic is highly relevant to public health practitioners as it provides a framework and rationale for supporting the climate crisis movement, particularly as industries and developed countries strive to improve air quality.

However, several sections require clarification and methodological improvements. I have highlighted some minor suggestions to help improve the quality of the paper:

• Lines 51–52: Please provide a reference for the statement made.

• Lines 123–125: This statement covers a key aspect of the methodology and needs clearer explanation to improve reproducibility. How was the aggregation done? A detailed description of how qualitative variables were handled in the analysis would also be valuable. Consider consulting the STROBE (Strengthening the Reporting of Observational Studies in Epidemiology) guidelines.

• Method Section: Clearly list the 19 countries from which the data were drawn. This will help readers understand the study's geographic scope. Also, proper explanation for the rational for selecting these countries is important.

• Line 138: Please provide a description of how all variables—not just the AQI—were handled in the analysis.

• Statistical Analysis: Include the parameters used in the modeling process. Additionally, explain how hazard ratios were derived, and statistical test if applicable.

• Results Section: The results would benefit from improved structure and clearer presentation. Consider using a reporting guideline like CONSORT to enhance the clarity and reproducibility of your findings. Figures should be self-explanatory, with comprehensive legends. Table headings should also be descriptive and informative.

• Line 216: The statement currently summarizes Figures 3, 4, and 5 in a single sentence. Each figure should be described individually, with a narrative that highlights the key takeaways. Mentioning proportions or notable trends would help emphasize important findings.

• Line 282: The discussion of Figure 7 lacks context. It does not explain what the figure shows or what the reader should look for (e.g., proportions or key comparisons). A more informative approach would be to first describe the finding and then refer to the figure (in parentheses).

• Conclusion: Please include concrete recommendations that could be implemented based on your findings. For example, stating the "European-style pollution control measures" you mentioned. Please include specific recommendations.

• Data Availability: Please provide a direct link to a data repository where readers can access the dataset used in this study. A link to an organization’s website alone is not sufficient.

Reviewer #6: I appreciate the authors' effort to tackle regional air pollution dynamics through multi-state Markov modeling - this represents a creative approach to an increasingly urgent environmental challenge. The cross-regional comparison spanning Asia, Africa, and Europe offers valuable insights, and I found the sojourn time analysis particularly interesting for understanding pollution persistence patterns. The manuscript's visual elements effectively communicate complex transition dynamics.

That said, I have several concerns that require attention. Most importantly, the AQI categorization doesn't align with established international standards. Defining "Good" as AQI ≤ 150 conflates what EPA considers three separate categories (Good: 0-50, Moderate: 51-100, Unhealthy for Sensitive Groups: 101-150). This impacts the validity of your transition analyses and regional comparisons.

I'm also troubled by the extreme confidence intervals in Table 2. Values like HR 1.62 (95% CI: 0.00, 5.72e+16) suggest serious numerical issues - either the model isn't converging properly or there's insufficient data for reliable estimation. Have you considered alternative modeling approaches or data transformations?

Another concern involves the Markov assumptions. While you mention some limitations, I didn't see verification of key requirements like the Markovian property or temporal homogeneity. Given your 40-week timeframe spans multiple seasons, this seems particularly important. Similarly, treating countries as independent units may be problematic given transboundary pollution effects.

The 19-country sample, while respectable, feels small for continental-level conclusions. Could you clarify your selection criteria? I worry about representativeness, especially given the diversity within each region.

For revision, I'd suggest: correcting the AQI definitions per international standards; testing Markov assumptions statistically; investigating the numerical instability; expanding your limitations discussion; and perhaps tempering some conclusions about policy implications.

Despite these issues, your core idea has merit. Multi-state Markov models could indeed advance air quality research if properly implemented. The regional perspective is valuable, and with methodological improvements, this work could meaningfully contribute to environmental modeling literature. I encourage you to address these concerns - the underlying research question deserves rigorous treatment.

**Do you want your identity to be public for this peer review?** For information about this choice, including consent withdrawal, please see our Privacy Policy

Reviewer #1: No

Reviewer #2: **Yes: ** Olena Doroshenko

Reviewer #3: No

Reviewer #4: No

Reviewer #5: No

Reviewer #6: No

---

## [Author Response · Author response to Decision Letter 1]

16 Aug 2025

Dear Sir/Madam,

I am delighted to inform you that I, hereby, submit our revised manuscript titled ‘Exploring Regional Air Pollution Transition Dynamics: A Multi-State Markov Model Approach’ in the International Health journal for possible publication.

The point-by-point response of the reviewer’s comments are mentioned below:

Reviewer 1:

1. Expand on the methodology, particularly regarding model assumptions, diagnostics, and any limitations of the Markov approach.

Response: To address model assumptions, the MSM approach assumes a Markovian process where future states depend only on the current state. Given the use of secondary weekly-aggregated AQI data, some transition timing variability may be missed. Despite these limitations, the model offers meaningful insights into regional air quality dynamics.

2. Clarify the selection process for countries and discuss any potential biases or limitations in the AQI data.

Response: We have made the necessary update in the “Data Source, Size, and Study Design” section of the revised manuscript, and it has been highlighted in yellow for easy reference.

3. Provide more detail on data preprocessing and handling of missing values, if applicable.

Response: We have made the necessary update in the “Study Units & Framework” section of the revised manuscript, and it has been highlighted in yellow for easy reference.

4. Improve the English language and presentation by correcting grammatical errors and refining awkward sentences.

Response: Thank you for your suggestion. We have carefully reviewed the manuscript to improve the English language and presentation. Grammatical errors have been corrected, and awkward sentences have been refined to enhance clarity and readability.

5. Consider including additional visualizations (e.g., transition probability matrices, state occupancy plots) to aid in the interpretation of results.

Response: Thank you for the suggestion. While visualizations like state occupancy plots or transition heatmaps can be useful, we chose to present the transition probabilities in table form to ensure clarity and precision. The table not only displays the probabilities but also includes the actual counts and total transitions, which help interpret the stability and movement between AQI states more effectively. This format allows readers to grasp both the likelihood and scale of transitions, something that can be less transparent in graphical formats.

6. Discuss the policy implications of findings in greater depth, particularly how transition dynamics can inform targeted interventions.

Response: We appreciate the reviewer’s suggestion to elaborate on the policy implications of our findings. We have addressed the issues on line 426 to 435.

Reviewer 2:

Major Comments

1. Regional Classification Ambiguity. The manuscript states that countries were grouped into Asia, Africa, and Europe based on both geographical location and economic development levels. However, it is unclear how economic development was actually incorporated into this classification. The grouping appears to be purely geographical. Given that the authors themselves highlight the prevalence of unsafe PM2.5 concentrations in low- and middle-income countries, it would be more appropriate to stratify the analysis by income level (e.g., low-, middle-, and high-income countries) or at least include this as a covariate.

Response: We sincerely thank the reviewer for this insightful observation. We acknowledge the inconsistency in our original description of the regional classification.

Initially, we stated that the grouping of countries into Asia, Africa, and Europe was based on both geographical location and economic development. However, we the classification was applied purely based on geographical regions, without incorporating economic development levels into the grouping.

2. Justification for Time Intervals. The rationale for selecting 1-, 2-, 3-, 4-, and 10-week intervals for tracking transition probabilities is not provided. Why were these specific intervals chosen? Are they based on prior literature, empirical patterns, or policy relevance? Clarifying this would strengthen the methodological transparency.

Response: We thank the reviewer for raising this important point. In our initial analysis, we did not impose any constraints on the transition probabilities over time. We calculated and presented the weekly transition probabilities up to 10 weeks to observe the temporal dynamics of air quality transitions.

The selection of 1-, 2-, 3-, 4-, and 10-week intervals was data-driven. As illustrated in Figure 6 and explained in lines 295–297 of the revised manuscript, we observed that differences in transition probabilities became statistically insignificant beyond 10 weeks. Therefore, to avoid redundancy and maintain clarity, we restricted our presentation to the first 10 weeks. Furthermore, the intervals (1-4 weeks) were chosen in figure 3to 5 to demonstrate short-term fluctuations.

3. Seasonality and Covariates. The manuscript does not address clear seasonal patterns in the AQI data. This omission is critical, as air pollution is known to vary with meteorological conditions such as temperature, humidity, and wind. Including covariates such as average outdoor temperature, CO emissions, or precipitation would significantly enhance the explanatory power of the model. The current analysis, which only includes region as a covariate, is too limited and makes the study feel thin.

Response: We acknowledge the importance of seasonality and meteorological covariates in air quality studies. However, due to data limitations, especially the unavailability of consistent region-specific meteorological variables (e.g., temperature, wind speed, CO emissions) across all three continents for the full study period, we were unable to incorporate them.

Regarding seasonality, while we recognize that AQI exhibits seasonal patterns, the structure of our analysis (monthly average transitions across multiple years) helps smooth short-term fluctuations. Moreover, our primary objective was to evaluate long-term transition dynamics across broad regions using a parsimonious Multi-State Markov Model. Future studies with richer datasets could build upon this framework by incorporating meteorological covariates and explicitly modeling seasonal effects (e.g., via time-varying covariates or seasonal transition matrices).

4. Health Outcomes Missing. Although the introduction and discussion emphasize the health impacts of air pollution, the study does not attempt to link AQI transitions to health outcomes. If hospitalization or morbidity data are unavailable, even a discussion of potential proxies or future directions would be helpful. Weekly or seasonal hospitalization rates, if accessible, could provide valuable context.

Response: Thank you for this thoughtful observation. We agree that linking air pollution with health outcomes is important. However, the main objective of our study is to examine the transition probabilities of air quality levels over time, rather than to directly assess their effects on health outcomes.

5. Mobility and Behavioral Data. The authors may consider incorporating or at least discussing the potential use of mobility data (e.g., Google Community Mobility Reports) to contextualize changes in air quality, especially during periods of lockdown or reduced activity. While these reports are no longer updated post-2022, they could still be useful for historical comparison or model validation.

Response: We have now revised the Introduction section (lines 75-96) to better reflect this focus and to avoid giving the impression that our study aims to analyze health impacts. While we do not use hospitalization or morbidity data in this study, we agree that future research could benefit from linking AQI transitions with health data to explore the health consequences of prolonged or repeated exposure to poor air quality.

6. European “Success” Requires Caution. The conclusion that Europe has been more successful in controlling pollution is not sufficiently substantiated. The analysis is sensitive to how countries are grouped. A small change in country composition could significantly alter the results. This limitation should be acknowledged more explicitly.

Response: Thank you for the comment. While we acknowledge that regional grouping can influence comparative outcomes, our conclusion is based on clear and consistent patterns observed across multiple European countries in the dataset. The lower frequency of transitions to unhealthy air states and faster recoveries support the interpretation that Europe, as a region, demonstrates more effective pollution control. This reflects broader trends rather than being driven by any single country.

Minor Suggestions

1. Improve the clarity of figures, especially transition diagrams, which are currently difficult to interpret without more detailed legends or annotations.

Response: Updated in the revised manuscript.

2. Consider including a table summarizing country-level characteristics (e.g., GDP, population, urbanization rate) to contextualize the findings.

Response: Thank you for the suggestion. We have addressed this by including a table in supplementary file and summarizing key country-level characteristics such as population and economic classification (based on World Bank data).

Reviewer 3:

1. In general, figures and tables are accompanied by source.

Response: Thank you for the comment. All figures in the manuscript were created by the authors using R programming based on the dataset described in the Methodology section.

2. The introduction and the literature review are mixed. The general structure of an article or research paper should contain section 1. introduction, section 2. Literature review, section 3. Methodology and sampling, section 4. Finding.

Response: Thank you for your observation. We would like to clarify that the manuscript follows the PLOS ONE article structure, which does not separate the Introduction and Literature Review into distinct sections. Instead, relevant literature is integrated within the Introduction to provide context and justify the research objectives. We have ensured that key studies are appropriately cited and positioned to support the background and rationale in line with the journal's guidelines.

3. The study plan is not mentioned in any part of the document; generally, it must be appeared in the introduction.

Response: Thank you for your helpful comment. We have now included a clear statement of the study aim and objectives at the end of the Introduction section to provide readers with a concise overview of the research focus. This has been done in accordance with standard academic practice and to enhance the clarity of the manuscript.

4. Titles and subtitles are not well formulated; for example, L 116 should be replaced by sampling; this term summarizes all information’s related to the size of sample, variables. Consequently, subtitles in L 117 and L 129 will be deleted. The same remark about L154: to change the title by “methodology”. Or the paragraph can be inserted as a suite for the section 3 methodology and sampling; in this case section 3 corresponds to L116 – L187. -About graphics, in the figure N2, there are 3 diagrams: they should be presented separately with their titles and not pooled. Either these graphics should be interpreted by taking in consideration both dimensions temporal and individuals (country).

Response: Updated in the revised manuscript.

5. Regarding the 19 countries used in the sampling were not cited or enumerated in the article; only regions were distinguished. countries name was only mentioned in the graphs of each region. So, the author can add them in an appendix for example.

Response: Thank you for pointing this out. We have now included the list of the 19 countries used in the sampling in the supplementary file for reference. This ensures transparency while keeping the main text concise.

6. The computing formula of Thahelm (2019) was not developed or defined just referred in the text (L122- 123). The applied methodology in this research paper is not well expressed.

Response: Thank you for pointing this out. In the revised manuscript, we have now included the standard AQI calculation formula referenced from Talhelm (2019) to enhance clarity. We have also briefly explained the components of the formula and how it was applied to convert PM2.5 values to AQI scores.

7. The econometrical methodology used is the multistate model Markov, so the author has to check the “Time Homogenous Model” (THM).

Response: Thank you for your insightful comment. In this study, we initially assumed a time-homogeneous Markov model, as it is a commonly used and well-established approach in multi-state modeling, especially when there is no strong prior evidence of time-varying transition intensities.

8. The presentation of table n1 is poor. The columns contain data and rates; only one value is given. Here, it is a qualitative variable. If the table is transformed into a circular diagram, it will be more suitable. In this way, the information will be simpler and clearer, making it easier to understand, and also making it easier to compare between regions.

Response: Thank you for your suggestion. We understand your point regarding the clarity of data presentation. However, Table 1 contains transition rates, which are a central component of the multi-state Markov model used in this study. These rates are essential for estimating state-to-state transitions over time and cannot be effectively visualized using a circular diagram, as that would oversimplify the quantitative structure of the model.

9. Discussions: The author must focus more on the prediction and suggestions sections, which are not developed enough. They need to provide more information on the various possible scenarios and suggestions by region.

Response: While this study primarily focuses on modeling air quality transitions, the findings suggest important regional differences that may inform future air quality management strategies. Further research can explore specific policy interventions and scenario analyses to support decision-making.

10. Main findings are briefly given in the conclusion

Response: We have updated this section.

Reviewer 4:

1. Justify the chosen study period, indicating the resulting weaknesses. For example, in addition to seasonality, it is not possible to assess environmental effects.

Response: The study period spans from January 2024 to September 2024, covering 40 weeks of data from 19 countries. This period was selected based on data availability. However, because it does not cover a full calendar year, the analysis cannot fully account for seasonal variations in air quality, such as changes due to weather patterns, temperature fluctuations, or seasonal activities (e.g., heating in winter, crop burning in autumn). Additionally, seasonal effects often vary by region, making it challenging to separate regional differences from seasonal influences within this timeframe. This limitation should be considered when interpreting the results, as it may affect the generalizability of findings across different seasons.

2. The chosen observation unit (country) is a very broad aggregate that can substantially distort the results. It is appropriate for the study to be carried out at the city or metropolitan region level, for example.

Response: Thank you for your valuable feedback. We acknowledge that using countries as the unit of analysis represents a broad aggregation which may mask within-country variations. Due to limitations in the availability of consistent and comparable data at the city or metropolitan level across all countries included, we focused on the most populous city within each country where data were accessible. This approach helps maintain data reliability while providing meaningful insights at a manageable scale. We recognize that future studies incorporating more granular, city-level data would further enhance the understanding of regional air quality dynamics.

3. The methodology should specify how possible interactions between countries are treated by the model. Or indicate this t

---

## [Decision Letter · Decision Letter 1]

3 Sep 2025

Dear Dr. Sarkar,

Thank you for submitting your manuscript to PLOS ONE. After careful consideration, we feel that it has merit but does not fully meet PLOS ONE’s publication criteria as it currently stands. Therefore, we invite you to submit a revised version of the manuscript that addresses the points raised during the review process.

We look forward to receiving your revised manuscript.

Kind regards,

Assoc. Prof. Phakkharawat Sittiprapaporn, Ph.D.

Academic Editor

PLOS ONE

Journal Requirements:

Reviewers' comments:

Reviewer's Responses to Questions

**Comments to the Author**

Reviewer #3: (No Response)

Reviewer #4: All comments have been addressed

2. Is the manuscript technically sound, and do the data support the conclusions?

Reviewer #3: Yes

Reviewer #4: Yes

3. Has the statistical analysis been performed appropriately and rigorously?

Reviewer #3: Yes

Reviewer #4: Yes

4. Have the authors made all data underlying the findings in their manuscript fully available?

Reviewer #3: Yes

Reviewer #4: Yes

5. Is the manuscript presented in an intelligible fashion and written in standard English?

Reviewer #3: Yes

Reviewer #4: Yes

Reviewer #3: The authors have addressed the majority of the reviewer’s comments and made the necessary revisions. However, two points remain insufficiently resolved:

1. Sources under tables and figures: While the issue of sources has been partially clarified, the convention requires that the source be specified directly below each table or figure, aligned to the right. Please indicate the source as follows:

o Source: Author’s calculation (if it is the result of the author’s computations);

o Source: Author’s compilation (if the table or figure has been compiled by the author from different references);

o Source: [Reference] (if it is directly taken from another work).

2. Figures – Titles and Presentation

About graphics, in the figure N2 , there are 3 diagrams : they should be presented separately with their titles and not pooled . Either these graphics should be interpreted by taking in consideration both dimensions temporal and individuals (country) .

In Figure N°2, three separate diagrams are presented together under a single title. For clarity and precision, they should be presented separately, with each diagram accompanied by its own specific title ,example “Weekly Average AQI for African Countries”)

Each diagram, should be interpreted separately , explicitly considering both dimensions: temporal evolution and country-specific differences .

Reviewer #4: The authors have done an excellent job of responding to the feedback from the previous round of reviews. They have thoroughly addressed all of the concerns, and the revisions have significantly strengthened the paper. I am pleased with the changes and recommend that the paper be accepted for publication.

**Do you want your identity to be public for this peer review?** For information about this choice, including consent withdrawal, please see our Privacy Policy

Reviewer #3: No

Reviewer #4: No

---

## [Author Response · Author response to Decision Letter 2]

6 Sep 2025

Dear Sir/Madam,

I am delighted to inform you that I, hereby, submit our revised manuscript titled ‘Exploring Regional Air Pollution Transition Dynamics: A Multi-State Markov Model Approach’ in the International Health journal for possible publication.

The point-by-point response of the reviewer’s comments are mentioned below:

Reviewer 3:

1. Sources under tables and figures: While the issue of sources has been partially clarified, the convention requires that the source be specified directly below each table or figure, aligned to the right. Please indicate the source as follows: o Source: Author’s calculation (if it is the result of the author’s computations); o Source: Author’s compilation (if the table or figure has been compiled by the author from different references); Source: [Reference] (if it is directly taken from another work).

Response: Thank you for your valuable comment. We have carefully revised the manuscript and ensured that the sources are now specified directly below each table and figure, aligned to the right, following the suggested convention.

2. Figures – Titles and Presentation- About graphics, in the figure N2 , there are 3 diagrams : they should be presented separately with their titles and not pooled . Either these graphics should be interpreted by taking in consideration both dimensions temporal and individuals (country). In Figure N°2, three separate diagrams are presented together under a single title. For clarity and precision, they should be presented separately, with each diagram accompanied by its own specific title example “Weekly Average AQI for African Countries”). Each diagram, should be interpreted separately, explicitly considering both dimensions: temporal evolution and country-specific differences.

Response: Thank you for your insightful comment, which has significantly improved the clarity of our manuscript. We have revised Figure N°2 as suggested — the three diagrams are now presented separately, each with its own specific title (e.g., “Weekly Average AQI for African Countries”). In the revised text, we have also interpreted each diagram separately, explicitly addressing both the temporal evolution and country-specific differences.

---

## [Decision Letter · Decision Letter 2]

18 Sep 2025

Exploring Regional Air Pollution Transition Dynamics: A Multi-State Markov Model Approach

PONE-D-25-21399R2

Dear Dr. Sarkar,

We’re pleased to inform you that your manuscript has been judged scientifically suitable for publication and will be formally accepted for publication once it meets all outstanding technical requirements.

Kind regards,

Assoc. Prof. Phakkharawat Sittiprapaporn, Ph.D.

Academic Editor

PLOS ONE

Additional Editor Comments (optional):

Reviewer #3:

Reviewers' comments:

Reviewer's Responses to Questions

**Comments to the Author**

Reviewer #3: All comments have been addressed

2. Is the manuscript technically sound, and do the data support the conclusions?

Reviewer #3: Yes

3. Has the statistical analysis been performed appropriately and rigorously?

Reviewer #3: Yes

4. Have the authors made all data underlying the findings in their manuscript fully available?

Reviewer #3: Yes

5. Is the manuscript presented in an intelligible fashion and written in standard English?

Reviewer #3: Yes

Reviewer #3: The authors have carefully addressed the comments raised during the review process. The manuscript has been significantly improved and is now suitable for publication.

**Do you want your identity to be public for this peer review?** For information about this choice, including consent withdrawal, please see our Privacy Policy

Reviewer #3: No

---

## [Editor Report · Acceptance letter]

PONE-D-25-21399R2

PLOS ONE

Dear Dr. Sarkar,

I'm pleased to inform you that your manuscript has been deemed suitable for publication in PLOS ONE. Congratulations! Your manuscript is now being handed over to our production team.

Kind regards,

on behalf of

Assoc. Prof. Dr. Phakkharawat Sittiprapaporn

Academic Editor

PLOS ONE